# Recent Advances in Chemically Modified Cellulose and Its Derivatives for Food Packaging Applications: A Review

**DOI:** 10.3390/polym14081533

**Published:** 2022-04-10

**Authors:** Zhuolun Jiang, To Ngai

**Affiliations:** Department of Chemistry, The Chinese University of Hong Kong, Shatin, N.T., Hong Kong, China; 1155140454@link.cuhk.edu.hk

**Keywords:** cellulose, biodegradable polymers, chemical modification, food packaging

## Abstract

The application of cellulose in the food packaging field has gained increasing attention in recent years, driven by the desire for sustainable products. Cellulose can replace petroleum-based plastics because it can be converted to biodegradable and nontoxic polymers from sustainable natural resources. These products have increasingly been used as coatings, self-standing films, and paperboards in food packaging, owing to their promising mechanical and barrier properties. However, their utilization is limited because of the high hydrophilicity of cellulose. With the presence of a large quantity of functionalities within pristine cellulose and its derivatives, these building blocks provide a unique platform for chemical modification via covalent functionalization to introduce stable and permanent functionalities to cellulose. A primary aim of chemical attachment is to reduce the probability of component leaching in wet and softened conditions and to improve the aqueous, oil, water vapor, and oxygen barriers, thereby extending its specific use in the food packaging field. However, chemical modification may affect the desirable mechanical, thermal stabilities and biodegradability exhibited by pristine cellulose. This review exhaustively reports the research progress on cellulose chemical modification techniques and prospective applications of chemically modified cellulose for use in food packaging, including active packaging.

## 1. Introduction

Plastics and common polymers made from fossil feedstocks have helped to build the modern world. However, ubiquitous discarded plastic contaminates the environment and has caused a serious microplastics problem, threatening the health of marine life, with associated risks for ecosystems and ultimately for human health, owing to plastic’s long durability over centuries in terrestrial and water environments [1,2]. Efforts to reuse and reduce plastic have proven insufficient, and most plastic waste accumulates in landfills or is released into the environment [3,4]. Food packaging materials account for approximately 26% of all plastic produced worldwide and play a vital role in preserving food throughout the distribution chain, owing to plastic’s water barrier capability, low cost, and light weight. However, approximately 90% of food packaging becomes waste after only one use [5,6]. With increased environmental concerns regarding sustainability and end-of-life disposal challenges, governments have proposed regulations such as limits or bans on single-use plastic bags and straws. Thus, there is an urgent need to transition from using plastic packaging materials to sustainable, biodegradable or compostable materials for food packaging [2,7,8,9,10,11,12,13]. Moreover, the ongoing COVID-19 pandemic has caused factory and transport disruptions worldwide, and much more food packaging material needs to be produced to meet the need for food preservation.

Cellulose refers to a class of natural carbohydrate polymers that are found in a virtually inexhaustible source of raw materials, such as plants, agricultural residues, shells of marine organisms, and microorganisms (Figure 1a) [14,15]. The plant cell wall of cellulose fibers contains different ratios of cellulose embedded as microfibrils in a matrix of hemicellulose, lignin, pectin, ash, and other extractives, depending on the origin of plants [14,16,17]. Cellulose is a class of linear, stereoregular, semicrystalline polysaccharides composed of repeated *β*-1,4-linked D-anhydroglucose units (AGU) with three hydroxyl groups (–OHs) per unit (Figure 1b) [18,19]. Compared with conventional petroleum-based polymers, such as polyethylene (PE), polypropylene (PP), and polystyrene (PS), cellulose-based materials have higher thermal stability, better film-forming capability, and better application qualities, such as a lower thermal expansion coefficient and improved mechanical strength and thermal stability at extreme temperatures [20,21,22]. Its intrinsic renewability and biodegradability make this material highly promising for use in various practical fields, e.g., packaging, straws, foams, flexible electronics, and tissue engineering materials, for the realization of sustainable product solutions and achieving a low carbon footprint (Figure 1c) [13,22,23,24,25,26,27,28,29,30,31,32,33].

Desirable food packaging should provide a sufficient barrier against grease, water vapor, and oxygen, maintain good mechanical strength, and continually release antimicrobial agents during food storage [34,35,36,37,38,39,40]. These properties will extend the shelf-life of packaged food by inhibiting microbiological growth and reducing gas and moisture exchange between the food and the surrounding environment to slow chemical and physical changes in food [35,41,42,43,44,45,46]. Cellulose fibers can be manufactured into wrapping films and containers with various shapes, owing to the multiple hydrogen bonds among fibrils [6,13,47,48]. Nanosized cellulosic fibers have a higher specific surface area than cellulose fibers, and their high hydrogen bonding formation ability allows the material to create a strong and dense network, which makes it difficult for molecules to pass through [49,50,51]. This property is beneficial for barrier applications, especially to prevent the passage of oxygen, which is highly useful for the food packaging industry [52,53,54]. The water vapor transmission rate (WVTR) and oxygen transmission rate (OTR) of cellulose-based packaging are two critical indices that should be reduced as much as possible to achieve sufficient food packaging barrier performance, and should be comparable with those of commercial plastic products, such as low-density polyethylene (LDPE) [15,55,56]. It is essential to increase the hydrophobicity of cellulosic packaging to maintain its strong barrier ability and to enable its use in extended applications, even under humid conditions or when in the presence of moist foods [57]. The inherent hydrophilic character of cellulose fiber results in moisture absorption and the swelling of the polymer, leading to a more porous structure and thus a reduction in the packaging barrier to moisture and oxygen and degraded mechanical properties caused by interfibrillar slippage [17]. However, current methods of reducing the high hydrophilicity of cellulose-based food packaging materials are usually not good for the environment, such as coating the surface of the cellulose-based material with a thin layer of wax, plastic (e.g., PE), or aluminum. These coatings, which are prevalent in the production of milk cartons and paper cups, hinder the biodegradability of the packaging, resulting in potential environmental contamination [17].

This review begins with a discussion of fundamental information regarding the natural origin of cellulose, fiber extraction methods, and the scale, crystallinity, and functionality of fibers. It then systematically describes and provides up-to-date information on the chemical modification techniques of cellulose and its derivatives, along with giving details about the degree of reaction, degree of substitution (DS), and packaging preparation methods. Moreover, the influence of chemical modification on cellulose properties is discussed, with a focus on the barrier properties (surface hydrophobicity, oil/water vapor/oxygen barrier) and mechanical and thermal properties of cellulose-based packaging. Routes for improving the packaging performance to mitigate the intrinsic drawbacks of certain chemical modifications that can reduce the mechanical strength and water vapor barrier are outlined. Safety and biodegradability issues are also briefly considered. To highlight value-added applications in food preservation, this review also gives several examples in which various categories of additives (plasticizers, antioxidants, antimicrobial agents) are loaded. Approaches to overcoming the gap between industrially required extrusion–melting processing and the highly crystalline nature of cellulose are also examined, with the anticipation that a new generation of high-performance processable cellulose bioplastics is achievable.

Given the abundance of -OHs and derived functionalities of cellulose, chemical modifications such as (trans)esterification, amidation, silylation, urethanization, polymer grafting and crosslinking have been investigated intensively in the past decades to enhance the hydrophobicity of cellulose materials [16,58,59,60,61,62,63,64]. This review focuses on esterification, transesterification, and crosslinking techniques. The modification techniques can be further divided into two categories depending on whether the reaction occurs on the -OHs of neat cellulose or elsewhere on functional groups of cellulose derivatives (Figure 2). Theoretically, the extent of cellulose covalent functionalization is represented by a DS of 0–3, corresponding to the average number of -OHs per cellulosic AGU.

## 2. Cellulose Sources, Extraction Methods, and Size, Crystallinity and Functionality of Fibers

Approximately 36 cellulose chains come together to form a basic fibrillar unit, known as an elementary fibril, which has a characteristic lateral dimension of 1.5–3.5 nm and a length of up to 100 nm. These nanosized cellulose chains are bound together into larger cellulose microfibrils (CMF, 10–100 nm with a length up to 50 μm) through both intra- and intermolecular hydrogen bonds. CMF further associates to form plant fibers with diameters of 10–50 µm and lengths of several micrometers, depending on the cellulose source. The hierarchical microstructure of wood cellulose fibers is shown in Figure 3a [65,66]. The three-dimensional arrangement of the cellulose chains leads to the coexistence of crystalline and amorphous regions within the cellulose fibers [67,68]. Different mechanical and chemical treatments for cellulose plants can produce various micro- and nanosized cellulosic products through transverse dissociation in the amorphous region, namely, CMF, cellulose microcrystals (CMC), cellulose nanofibrils (CNF), and cellulose nanocrystals (CNC) (Figure 3b) [16,17,19]. These treatments lead to cellulose fibers with differing crystallinities, fiber sizes, and functionalities [16,69,70]. The traditional mechanical production of cellulose fiber mainly includes refining and high-pressure homogenization, microfluidization, and grinding [18,67,71]. CNC, also known as cellulose whiskers or rodlike cellulose (3–10 nm in diameter and 100–250 nm in length), is usually obtained via sulfuric acid hydrolysis of the amorphous regions of cellulose [72,73]. However, the introduction of charged sulfate groups compromises the thermostability of CNC, which can be a serious drawback for food packaging applications [74]. Bacterial nanofibrils (BNF) are composed of very pure cellulose nanofiber networks produced from low-molecular-weight sugars and alcohols through bacterial synthesis and have ultra-long nanofibers with diameter of 20–100 nm [26,75]. CMF, CMC, CNF, CNC, and BNF have emerged as key components for the production of cellulose-based food packaging. The first section of this review describes several studies in the last 10 years that have investigated fundamental aspects of the raw sources of cellulose, the production methods of micro/nanosized cellulose fibers, fiber size and crystallinity, and corresponding functionalities of the cellulose anhydroglucosic backbone (Table 1).

## 3. Esterification

The reactivity of acylation agents (acylants) has provided a broad spectrum of strategies for producing cellulose esters. The esterification of cellulose using carboxylic acids as acylants has low reactivity without a catalyst but can be activated by adding a strong acid such as sulfuric acid. However, during acidic catalysis, cellulose hydrolysis generally occurs simultaneously, which causes the degradation of glucosidic bonds. A good acylant alternative is to replace carboxylic acid with anhydride or acyl chloride, which possesses activated acyl moieties, in combination with a tertiary base (e.g., pyridine, triethylamine, 4-dimethylaminopyridine [DMAP]) [19,85]. Employing acyl chlorides and anhydrides for esterification leads to the formation of hydrochloric acid or a carboxylic acid, respectively, which are involved in the acidic hydrolysis of cellulose chains; this process can be alleviated by adding a base to the reaction medium to neutralize the progressive formation of acid products. The added base also has the advantage of being a catalyst, as bases can form reactive intermediates that drive the reaction forward [19,86].

### 3.1. Esterification with Acylants Bearing a Short Substituent Chain (C2–C6)

Cellulose esters have been used in commercial processes for decades. These materials mainly consist of cellulose with substituent lengths of no more than four carbon atoms. Such cellulose esters with short substitution chains (CESs) are generally produced by esterification using anhydrides in the presence of sulfuric acid [19,87]. Cellulose acetate (CA) is a commonly used CES and is described as the first organic cellulose ester, synthesized more than 150 years ago from wood pulp by Paul Schutzenberger [15]. Other CESs, including cellulose butyrate (CB), cellulose acetate propionate, and cellulose acetate butyrate (CAB), are also common commercial cellulose derivatives and are widely applied as components of coatings, paints, inks, membranes, and filters. These coatings are highly resistant to heat, UV, and moisture [88].

#### 3.1.1. Synthesis of CESs with and without Solvents

CESs can be synthesized using different types of acylants with and without organic solvents in the reaction system. Jonoobi et al. adopted a solventless method to synthesize CA using acetic anhydride (or acetic acid) as the acylant and dispersion medium with pyridine (or sulfuric acid) as a catalyst to modify kenaf CNF, CNC, and wood filter pulp. Under optimal reaction conditions obtained by adjusting the temperature and reaction period, the DS of CA was relatively low, and the water contact angle (WCA) of CA was increased to 113°, demonstrating the effectiveness of the hydrophobic modification [76,79]. Ramírez et al. also performed solventless esterification between BNF and acetic or propionic acid (as dispersant) at 120 °C for 1–8 h using nontoxic biobased tartaric acid as a catalyst. The acetylated BNF (DS = 0.45) and propionized BNF (DS = 0.23) samples did not absorb water and remained floating on the water surface, while the pristine BNF sample absorbed water and immediately sank to the vial bottom. The decomposition onset temperature (T_d-onset_, weight loss of approximately 5% upon heating) of the esterified samples was found to increase with the DS. The acetylated and propionized BNF showed higher thermal stability (T_d-onset_ = 275–330 °C) than that of pure BNF (225 °C) [77]. With toluene as a solvent, Singh et al. conducted esterification between CMF and propionic anhydride at 60–100 °C for 0.45–5 h with pyridine as a catalyst, obtaining CMF propionate with a DS of 0.34–2.56 and a WCA of up to 121° [62].

Many studies have used large quantities of anhydride, acyl chloride, or organic solvents (e.g., dimethylacetamide [DMAc], chloroform, toluene) for extraction or in various steps of the reaction. However, the reliance on these organic solvents is undesirable because they are toxic, flammable, and/or volatile, cause air pollution, are hazardous to human health, and lead to difficult recoveries. To avoid these noxious substances, an effective water-based method was developed to endow the cellulose surface with a high carboxyl content through the esterification of cellulose with oxalic acid. The obtained oxalic acid modified CMF (OCMF) aqueous solution was then deposited onto filter paper through vacuum filtration to prepare the composite paper. The T_d-onset_ of the OCMF sample decreased by approximately 40 °C compared with that of pristine CMF. Notably, while the OCMF deposited on the filter paper was still hydrophilic (WCA = 63°), its WVTR and grease-stained area markedly decreased from 694 to 123 g/m^2^ 24 h and from 91% to 0%, respectively, compared with pristine filter paper [89]. Swatlosk et al. reported a novel type of “green” solvents called ionic liquids, which show very low vapor pressure and the possibility for recycling and reuse after the reaction [90]. Ionic liquids have attracted considerable attention over the past two decades for the homogeneous modification of cellulose, despite their high cost [80,83,91,92]. Missoum et al. used acetic, butyric, iso-butyric, and hexanoic anhydrides to esterify CNF in an ionic liquid 1-butyl-3-methylimidazolium hexafluorophosphate [bmim][PF_6_] at 100 °C for 2 h. The WCAs of CNF acetate, CNF butylate, CNF iso-butylate, and CNF hexanoate films were found to be 56°, 99°, 100°, and 105°, respectively. The ionic liquid was successfully recycled without any impurities or by-products formed during the reaction through two cycles of washing in sodium hydroxide solution and further purification through two cycles of extraction with diethyl ether [83].

#### 3.1.2. CESs in Food Packaging Applications

In the past 10 years, CES-coated/coupled papers and CES self-standing films have become commonly used as food packaging materials. CA film displayed a higher WCA (83°) and lower WVTR (167 g/m^2^ 24 h) than an unmodified CMF film (WCA = 41, WVTR = 234 g/m^2^ 24 h), and the OTR of the CA film (7.5 mL m^−2^ day^−1^) was higher than that of the original CMF film (4.2 mL m^−2^ day^−1^), demonstrating that acetylation could weaken the oxygen barrier of the cellulose film. Furthermore, the tensile strength (σ) of CA films was slightly reduced compared with the original CMF film [93]. CA can be composited with paperboard to produce a material that can be made into food trays. When the active filler-layered double hydroxide (LDH) intercalated with antimicrobial 4-hydroxybenzoate anion (LDH-HB) was embedded in food-grade resin and coated onto CA-coupled cardboard, the material showed a high antibacterial effect and was suitable for food contact. This packaging was found to effectively preserve cooked tomato pasta for up to 30 days at 4 °C [5]. Transparent and flexible butyrated cellulose nanocrystal (Bu-CNC) films and coatings (DS = 2.1) showed an enhanced WCA of 92°. The Bu-CNC finally formed a dry, white, spongy material on the water surface when CNC formed a clear transparent suspension in water. The T_d-onset_ of Bu-CNC was nearly 50 °C higher than that of the original CNC (257 °C). When Bu-CNC was coated onto mung bean seeds, there was no sprout formation, while CNC-coated seeds showed obvious sprouting, demonstrating Bu-CNC as a promising waterproofing coating for food [94]. Ioelovich et al. prepared various CESs with high DS values of 2.7–3, including CA, cellulose propionate (CP), CB, and CAB. The CESs were made into films and coatings for paperboard. The hydrophobicity of the cellulose esters increased with substituent chain lengths from C2 to C4. CAB showed high hydrophobicity that was comparable to that of CP. The 10 wt% CES-coated paper became completely resistant to water and oil, while the original paper had no effective barrier against water or oil. The water absorption (WA) of the CB- and CAB-coated paper was reduced to 0.1 g/m^2^ and 0.2 g/m^2^, respectively, compared with 98 g/m^2^ for pristine paper. These CES films displayed a high elongation at break (ε) (43–48%). When immersed in deionized (DI) water for 24 h, the σ of the CAB and CB films (~33 MPa) remained nearly unchanged compared with their dry counterparts [95]. Deng et al. reported an efficient and easily industrialized method for the synthesis of cellulose hexanoate (CH) through the acylation reaction between CMF and hexanoyl chloride in dimethyl formamide (DMF) activated by mechanical ball milling at room temperature (RT). The CH films had good transparency compared with the white and opaque pristine CMF film. At the optimal milling time and acylant dose, the CH films (DS = 0.25–0.45) displayed lower WA (4%), a higher WCA (113°), and a lower WVTR (101 g/m^2^ 24 h) than the pristine CMF film (70%, 20°, 607 g/m^2^ 24 h). Furthermore, the acylation reaction (3 mL of acylant) combined with ball milling treatment (6 h) endowed the CH film with a markedly increased σ and ε (140 MPa, 21.3%) compared with the control CMF film (7 MPa, 4.4%). However, the T_d-onset_ of CH markedly decreased with increasing DS, but remained above 280 °C [96]. Esterification of cellulose using C2–C6 acylants, reaction degree, and the associated packaging forming methods are listed in Table 2.

#### 3.1.3. CES-Based Food Packaging Containing Various Additives

Additives such as plasticizers, antioxidants, and antimicrobials are often incorporated to improve the specific properties of CES films. Cinnamaldehyde (CIN) or green-synthesized silver nanoparticles (G-AgNPs) were incorporated into CA coatings or self-standing films, greatly increasing the antimicrobial performance of the CA film or paper substrate. The composite packaging showed low cytotoxicity and high antioxidation ability, and beneficial properties were promoted, such as an increased barrier to oil, water, water vapor, oxygen, and increased mechanical strength. In addition, the food shelf-life of beef was prolonged by 4–5 d at 4 °C [37,97]. As pure CA films are very brittle, plasticizers are generally incorporated into the CA film to make it flexible by increasing the mobility of the polymer chains, which greatly influences the Young’s modulus (E), σ and ε of the CA packaging. Poly(ethylene glycol) (PEG) and carotenoids have been shown to be effective plasticizers for CA films due to their ability to enhance film elongation. The addition of cetrimonium bromide (CTAB)-modified montmorillonite (CTAB-MMT) to PEG-CA film led to good antimicrobial performance with no cytotoxicity. With the addition of 3 wt% CTAB-MMT, the nanocomposite film exhibited the most improved WVTR (5.84 g/m^2^ 24 h) and σ (40.9 MPa) compared with the PEG-CA film, as the incorporation of clay layers into the polymer matrix creates a tortuous path that decreases water vapor diffusion through the polymer matrix [98]. The incorporation of carotenoids (lycopene, norbixin and zeaxanthin) into CA films protected sunflower oil and vitamin B_2_ from light oxidation. The films containing norbixin showed the highest barrier to UV-Vis light. The σ was increased from 65.3 for the CA film to 82.6, 105.6, and 87.4 MPa for norbixin-, lycopene-, and zeaxanthin-loaded films, respectively. A higher concentration of norbixin contributed to a higher water vapor permeability (WVP) (0.1 wt%: 0.035 g × mm m^−2^ h^−1^ kPa^−1^), which was associated with its hydrophilic character, while hydrophobic zeaxanthin and lycopene led to a decreasing trend in the WVP (0.1 wt%: 0.023 and 0.022 g × mm⋅m^−2^ h^−1^ kPa^−1^, respectively) compared with 0.032 g × mm⋅m^−2^ h^−1^ kPa^−1^ for the original CA film. The film containing 0.1% zeaxanthin showed decreased thermal stability (T_d-onset_ = 200 °C), which was approximately 50 °C lower than that of the pure CA film and the films blended with the other antioxidants [99,100]. C2–C6 CES materials used in food packaging applications are listed in Table 3.

### 3.2. Esterification with Acylants Bearing Medium (C8–C12) or Long Substituent Chains (>C12)

Grafting long substituent chains onto the cellulose surface greatly affects the hydrophobicity, barrier, mechanical strength, and thermal properties of cellulose. Generally, surface hydrophobicity (indicated by the WCA and WA), water vapor resistance (indicated by the WVTR), and elasticity (indicated by the ε) increase with the length of the substituent chain. The oxygen barrier (indicated by the OTR), σ, E and thermal stability (indicated by T_d-onset_) decrease with the length of the substituent chain.

#### 3.2.1. Influence of Substituent Length and DS on Surface Hydrophobicity and WVTR of Cellulose Esters with Medium (CEMs) and Long Substitution Chains (CELs)

Grafting medium or long substituent alkyl chains onto cellulose can effectively enhance the WCA and lower the water vapor transport of cellulose packaging (Figure 4). One study modified CNF using a polymer anhydride (maleated styrene block copolymers, *M*_n_ = 230,000) as an acylant, conferring CEL with a WCA of up to 130° [101]. Another study revealed the effectiveness of long alkyl chains for lowering the WVTR of cellulose film. Tall oil fatty acid (TOFA) is a side-stream of the pulping industry from coniferous trees. TOFA-based C18-CEL (DS = 2.5–2.9) film showed a strongly decreased WVTR of 22–43 g/m^2^ 24 h, which was similar to 20 g/m^2^ 24 h for LDPE [102]. In fact, under the same reaction period and temperature conditions, acylants with higher substituent lengths had lower DS values for cellulose esters. For example, with increasing substituent lengths of acylants from C2 to C12, the cellulose esters had a decreasing trend in the DS; from 1.23 to 0.64 for C2-CES to C12-CEM [103].

A higher DS of cellulose esters always results in a greater WCA and water vapor resistance. Zhang et al. synthesized C18-CEL0.3 (DS = 0.3) and C18-CEL3 (DS = 3) through esterification between CMC and stearoyl chloride. The C18-CEL3 film did not show significant WA, and its WCA increased to 110°, slightly higher than 102° for the C18-CEL0.3 film. The WVP of the C18-CEL3 film (5 × 10^−12^ g m^−1^ s^−1^ Pa^−1^) was much lower than that of the C18-CEL0.3 film (75 × 10^−12^ g⋅m^−1^⋅s^−1^ Pa^−1^), and both were considerably lower than 158 × 10^−12^ g m^−1^ s^−1^ Pa^−1^ for the neat CMC film [104]. Another report found that C6-CES film with a low DS (1.3) had a low WCA of 66°, with higher WCAs of 95–118° at higher DS values (1.6–2.9). This trend was similar for C16-CEL films, which showed a low WCA of 87° at DS = 0.8 and higher WCAs of 100–123° at DS = 1.6–2.6 [105,106]. At similar DS values, cellulose esters with longer substituent lengths have higher WCAs and water vapor barriers in the resulting cellulose ester packaging. C16-CEL film with a low DS of 0.9 showed a lower WVP (1.6 cc × mm m^−2^ d^−1^ kPa^−1^) compared with 6 cc × mm m^−2^ d^−1^ kPa^−1^ for a C6-CES film with a higher DS (1.3) [106]. The WVP of fully substituted cellulose esters (DS = 2.9–3) films was reported to decline from 32 to 4.5 × 10^−12^ g m^−1^ s^−1^ Pa^−1^ with increasing substituent length from C8 to C18 [107]. The WVTR (100–300 g/m^2^ 24 h) of C16-CEL (DS = 1.6–2.6)-coated paperboard was found to be considerably lower than that of C6-CES (DS = 1.6–2.8)-coated paperboard (400–1020 g/m^2^ 24 h) [105].

Self-standing films and paperboards are typically made through hot-pressing [107] or solution casting [102,106,107], and some studies have investigated the effects of coating techniques, such as spray-coating [108], bar-coating [105,109], or brush-coating [110] on the WCA and WVTR of cellulose ester-coated paperboards. A spray-coated ethanol suspension of C11-CEM (DS = 0.62) on filter paper conferred the coated paper with superhydrophobicity (WCA = 152°) due to the formation of claw-like bulges on the surface. However, the WVTR showed no change in comparison with that of uncoated filter paper because the micro/nanosized C11-CEM particles could not totally cover the pores of the underlying paper [108,109]. When a C11-CEM (DS = 2.75) solution immobilized of the antibacterial poly(hexamethylene guanidine hydrochloride) (PHGH) and 3-mercaptopropionic acid (MPA) was bar-coated on paper, the WCA and WVTR of the coated paper gradually decreased from 117° to 101° and from 441 to 192 g/m^2^ 24 h, respectively, with increasing coating thickness due to the markedly increased surface smoothness [109]. The number of coating sides also has a significant effect on the water vapor barrier of cellulose ester-coated paper. Balasubramaniam et al. investigated the effect of whole-paper immersion acylation and one-sided surface acylation on the hydrophobicity and water vapor barrier properties of premade CNF films. Lauroyl chloride, palmitoyl chloride, and stearoyl chloride were used as the acylants. For the one-sided acylation method, acyl chlorides were applied with a brush onto one surface of pyridine (100 °C)-swollen CNF film, and the films were allowed to stand overnight at RT. For the whole-paper immersion acylation method, CNF films were immersed in pyridine at 100 °C for 30 min, after which fatty acid chlorides were added and acylated for 90 min; the films were then dried. Immersion modification resulted in a higher DS (0.91–1.8) than the one-sided modification (0.37–0.55). The WCA of the one-sided-acylated CNF film was improved to 105–121°, and the WCA of the immersion-acylated CNF film was improved to 112–114°. The films formed through immersion modification showed a strongly enhanced water vapor barrier property (WVP = 0.006–0.021 ng s^−1^ m^−1^ Pa^−1^), while the one-sided modified films did not show any decrease in WVP compared with that of unmodified CNF films (WVP = 0.057 ng s^−1^ m^−1^ Pa^−1^) [110]. However, the WVTR values of the above cellulose ester films and coated paperboard were still much higher than that of commercial LDPE (20 g/m^2^ 24 h) and LDPE-coated paperboard (11 g/m^2^ 24 h) [102,105], although their surface hydrophobicity and water barrier property were considerably enhanced compared with unmodified cellulose film and paper.

#### 3.2.2. Influence of Substituent Length on OTR of CEMs and CELs

The oxygen barrier capacity of cellulose ester packaging can be seriously compromised by the disruption of the crystalline region, especially for CEL packaging, resulting in the acceleration of microbial activity and food spoilage [111]. Modified TOFA-acylated cellulose films (DS = 2.5–2.9) showed greatly decreased WVTRs of 22–43 g/m^2^ 24 h. However, these films could not be used as an oxygen barrier, because the concentration of oxygen passing through the films was unacceptably high, indicating that CELs function well as water vapor barriers but are not effective as oxygen barriers [102]. Films made from fully substituted cellulose esters (C8–C18) (DS = 2.9–3) showed a declining WVP from 32 to 4.5 × 10^−12^ g m^−1^ s^−1^ Pa^−1^ with an increasing length of the substituent chain. However, their oxygen permeability (OP) increased from 0.24 to 1.3 cm^3^ m^−1^ d^−1^ bar^−1^ with increasing substituent chain length, revealing that longer substituent chains caused poorer oxygen resistance [106,107].

#### 3.2.3. Influence of Substituent Length and DS on the Mechanical Strength of CEMs and CELs

Cellulose films have inferior mechanical strength compared with conventional plastics such as LDPE, which shows σ = 7–16 MPa, E = 102–240 MPa, and ε = 100%–800%. Grafting long substituent alkyl chains onto cellulose can further decrease the mechanical strength of the packaging, as hydroxyl bonding among cellulose fibrils is destroyed (Figure 5a,b).

At similar DS, cellulose ester packaging with increasing substituent lengths has decreased mechanical strength. The C8-CEM (DS = 1.3) film exhibited a similar E (380 MPa), higher σ, and markedly higher ε (19 MPa, 90%) compared with C18-CEL (DS = 1.3) films (E = 397 MPa, σ = 10.4 MPa, ε = 10.7%) [104,106]. The films of fully substituted C8–C16 cellulose esters (DS = 2.8) were ductile, except for the C18-CEL film obtained from stearoyl chloride, which was extremely brittle. The ε of C8–C16 cellulose ester films decreased significantly from 100% to 40% with increasing substituent chain length, while all had a low σ, ranging from 4.8 to 7.1 MPa [112]. A higher DS in cellulose ester films results in lower mechanical strength. C18-CEL (DS = 0.3, 3) films had a lower mechanical strength than that of unmodified CMC film (σ = 169.2 MPa, E = 7230 MPa, ε = 15.9%), but C18-CEL0.3 films showed a much better mechanical strength (σ = 28.5 MPa, E = 1118 MPa, ε = 12.7%) than C18-CEL3 films (σ = 5.5 MPa, E = 286 MPa, ε = 2.5%) [104]. C12-CEM (DS = 0.9) film had a significantly higher σ (35 MPa) than that of higher substituted C12-CEM films (DS = 1.9–2.8) (σ = 1.8–2.5 MPa) [106,113].

To overcome the serious degradation in mechanical strength of CEM and CEL self-standing films and paperboard, coating technology has been frequently adopted to maintain the original mechanical strength of the neat cellulose packaging (Figure 6). One study reported that C11-CEM-coated film and paperboard displayed unchanged mechanical strength, as revealed by a slightly increased ε and nonsignificant change in tensile index [109]. Another study reported that, when solutions of C6-CES and C16-CEL with DS values of 1.6–2.6 were bar-coated on paperboard, the σ of the coated paper was slightly enhanced to 12.5 MPa, compared with 11.8 MPa for the uncoated paperboard [105]. Balasubramaniam et al. investigated the effect of chemical coating method (immersion acylation or one-sided surface acylation) on the mechanical properties of premade CNF films. The mechanical strength of the one-sided modified C12–C18 CNF film (specific σ = 51–72 MPa g^−1^ cm^3^, specific E = 2.4–2.8 GPa g^−1^ cm^3^) was comparable to that of the original CNF film, while those of the C12–C18 films prepared from the immersion method were much weaker (10–23 MPa g^−1^ cm^3^, 0.4–1.6 GPa g^−1^ cm^3^) [110]. Sehaqui et al. immersed CNF wet cake into linear or cyclic C2–C16 anhydride liquid for 2 d, followed by hot pressing treatment. The C2-CES nanopaper (DS = 0.38) had the highest storage modulus, and the C16-CEL nanopaper (DS = 0.1) displayed the lowest storage modulus. However, the wet stability of the cellulose ester nanopapers progressively increased with the length of grafted moieties from C2 to C16. The C16-CEL nanopaper had the highest wet strength ratio (32%) of its initial dry strength, which constituted a 7-fold improvement compared with the reference CNF nanopaper [114]. Esterification of cellulose using C2–C20 acylants, reaction degree, and the associated packaging forming methods are listed in Table 4. C2–C18 cellulose ester materials used in food packaging applications are listed in Table 5. And the influence of packaging manufacturing technologies on properties of cellulose ester-based packaging (DS = 0.2–2.8) are listed in Table 6.

#### 3.2.4. Thermoplasticity of CELs

Cellulose has not yet achieved wide industrial acceptability for thermal processing; for example, it cannot undergo extrusion melting. The lack of thermal processing has led to increased production costs and times for cellulose-based packaging. One prerequisite for a polymer material to be suitable for melt processing is a broad temperature window, which means that the polymer’s melt temperature (T_m_) needs to be far lower than its T_d-onset_ [105]. The thermoplasticity of cellulose esters with substituent chains from C6–C16 has been reported in several studies, and is enabled by the internal plasticization effect of the long alkyl chains [84,103,110]. These studies have found that C6–C16 cellulose esters with low or moderate DS melt at 160–230 °C, but it has been difficult to measure their exact T_d-onset_ [105,106].

Generally, the T_d-onset_ of cellulose esters decreases with increasing substituent length [103]. The T_d-onset_ of fully substituted cellulose esters (C2–C18) has been reported to be 250–330 °C [102,117,118]. Fully substituted C8–C18 cellulose esters (DS = 2.7–3) are a class of thermosensitive polymers with a distinct T_m_ lower than 100 °C, but exhibiting a typical melting phenomenon (a thermoplastic feature) at relatively low heat, conferring the cellulose ester packaging with useful thermal responsiveness and processibility, greatly broadening its value-added application potential [104,105,106]. When the substituent length reaches C12 or higher, cellulose tri-esters are able to crystallize and form ordered regions via their side-chains. The T_m_ and heat of fusion of side-chain crystals increase with increasing side length from C12 to C20, which is attributable to an increase in side-chain crystal thickness as more methylene units participate in crystallization at longer side-chain lengths [87]. Sealey et al. (1996) found that fully substituted C12-CEM and C14–C20 CELs (DS =2.8–2.9) showed linearly increasing T_m_ values from −15 °C for C12-CEM to 55 °C for C20-CEL [87,113].

Zhang et al. found that fully substituted C18-CEL film displayed a distinct T_m_ at 55 °C, while no T_m_ was identified for the C18-CEL0.3 film. Reversible changes in C18-CEL3 film volume were observed when varying the temperature; these changes were caused by the construction and destruction of crystalline regions consisting of stearoyl moieties [104]. Geissler et al. prepared a transparent and thermally responsive cellulose film to controllably release rhodamine B molecules. C18-CEL (DS = 2.95) was synthesized through esterification between CMC and stearoyl chloride in pyridine. By dip-coating hydrophilic or hydrophobic silicon wafers into a C18-CEL toluene solution, ultrathin films were formed with thicknesses of 8–360 nm. The C18-CEL film was thermally responsive at temperatures above or below its T_m_ (56 °C). When the temperature was higher than 56 °C, the film melted and released the incorporated active agents (i.e., rhodamine B molecules), and the release of active molecules could be stopped by decreasing the temperature [119]. Crépy et al. synthesized fully substituted cellulose esters (DS = 2.8) using acyl chlorides from C8 to C18. The T_m_ values of the cellulose esters slightly decreased with increasing side-chain length from 58 °C for C8-CEM to 44 °C for C18-CEL. The results indicated that the cellulose esters organized into a layered structure in which the cellulosic backbones were arranged in a plane, and the flexible side-chains were fully extended and perpendicular to the cellulosic backbone. Furthermore, when the fatty chain length exceeded 12 carbon atoms, a portion of the alkyl chains crystallized into a hexagonal packing [112].

## 4. Transesterification

### 4.1. Transesterification with Vinyl Esters as Acylants

Many types of vinyl esters can be easily synthesized from the corresponding carboxylic acid. Transesterification is a relatively mild acetylation method, which uses acetate ester as an innocuous acyl donor [91]. Applying vinyl esters as transesterification agents in the presence of a catalyst has been commonly used for the formation of cellulose esters in the past 10 years [88,91,92,120]. The acylation approach based on the transesterification of vinyl esters is particularly attractive because it proceeds with high efficiency and without releasing acidic by-products. Even if the transesterification reaction is reversible, the vinyl alcohol leaving product is unstable and immediately tautomerizes to acetaldehyde, which is volatile and can be easily removed from the reaction system; this removal drives the reaction toward the formation of the expected esters, although the small-molecule aldehydes are toxic [88,120].

Brand et al. performed transesterification between CNC and vinyl acetate in dimethyl sulfoxide (DMSO) or DMF under microwave heating at 80–100 °C for 15–240 min, with K_2_CO_3_ as a catalyst. The fraction of accessible -OHs (*N*_OHs_) on CNC was calculated to be 16.7 mol%. The impacts of solvent, temperature, and the ratio of vinyl acetate to *N*_OHs_ on the reaction efficiency were investigated. The acetyl content as a percentage of total -OHs on CNC reached 14.4% in DMF and 12.8% in DMSO. The reaction was more efficient in DMSO than in DMF, owing to the better dispersion of CNC in DMSO. Increasing the molar ratio of vinyl acetate to *N*_OHs_ from 1.5 to 3 also significantly increased the reaction efficiency [120]. Cao et al. synthesized CA, CP, and CB, with high DS values of 2.14–2.34 through the transesterification between CMC and vinyl acetate, propionate, or butyrate in various solvents at 100 °C for 5 min with LiOH, NaOH, or KOH as a catalyst. Almost all reactants were soluble in DMSO, while considerable crystal precipitation was observed in DMF, DMAc, pyridine, and *N*-methylpyrrolidone. DMSO was thus identified as the most suitable solvent for this reaction. NaOH and KOH showed high catalytic activity in this transesterification reaction, and nearly all of the crystals disappeared within 5 min. In contrast, LiOH did not show sufficient catalytic activity. Therefore, DMSO, NaOH, and KOH were found to be suitable for the synthesis of CESs in this study. No T_m_ could be identified from the differential scanning calorimetry curve of CA, but T_m_ values could be identified from the curves of CP (234 °C) and CB (182 °C) [88]. Chen et al. used vinyl acetate as a transesterification agent to prepare CA from six cellulose raw materials (CMC, cotton linter pulp, wheat straw pulp, bamboo pulp, bleached softwood sulfite-dissolved pulp, and bleached hardwood kraft pulp (HP)) in DMSO at 100 °C for only 15 min, with NaOH as catalyst. The CA fibers prepared from the cellulose raw materials with the highest viscosity-average degree of polymerization (DPv) exhibited the lowest DS. CA from CMC had the highest DS (2.94) due to CMC having the lowest DPv, while CA from HP had the lowest DS (2.55), as a result of HP having the highest DPv, indicating that cellulose with longer chains has lower transesterification reactivity. The mechanical strength of CA increased with increasing DPv, and the thermal stability of CA decreased with the increasing DPv of cellulose. CA fibers showed higher thermal stability (T_d-onset_ = 290–320 °C) than unmodified cellulose (T_d-onset_ = 280 °C) [91]. Cao et al. performed transesterification between CMC and C2–12 vinyl esters in DMSO at 100 °C for 5 min with NaOH as catalyst. The DS values of the obtained C2–12 cellulose esters showed a decreasing trend from 2.02 to 0.76 with increasing substituent chain length, indicating that vinyl esters with longer aliphatic chains have lower transesterification reactivity. The T_d-onset_ values of the cellulose esters (320–333 °C) were higher than that of pristine CMC (T_d-onset_ = 314.5 °C), except for C2-CMC (312 °C) [121]. Wen et al. obtained C12-CEM through transesterification between CMC and vinyl laurate in the cosolvent of ionic liquid 1-allyl-3-methylimidazolium chloride [amim]Cl and DMSO at 70–120 °C for 1–6 h, with 1,8-Diazabicyclo[5.4.0]undec-7-ene (DBU) as a catalyst. The DS of C12-CEM increased from 1.47 to 2.41 when the reaction temperature was increased from 70 °C to 120 °C. At 120 °C, prolonging the reaction time from 1 to 6 h increased the DS from 1.97 to 2.74. The DS increased from 1.80 to 2.62 when the molar ratio of vinyl laurate to AGU was increased from 3:1 to 9:1, and the WCA of C12-CEM was increased to 96–121°. The E decreased from 73.28 MPa to 33.65 MPa and the ε increased from 50% to 116% upon increasing the DS from 1.97 to 2.74, suggesting that the cellulose films became more ductile after the introduction of more aliphatic chains. Increasing the DS of C12-CMC film from 1.97 to 2.41 resulted in an improved σ from 4.2 to 5.9 MPa, while further increasing the DS to 2.74 led to a decrease in σ to 5.1 MPa. C12-CEM displayed higher thermal stability (T_d-onset_ = 314–340°) than the original CMC (T_d-onset_ = 295°) [118].

### 4.2. Transesterification with Plant Oils as Acylants

Effective cellulose hydrophobization strategies have been demonstrated, based on applying activated acid derivatives (i.e., acyl chlorides or anhydrides) in stoichiometric amounts or excessive vinyl esters [91]. These treatments lead to cellulose with increased hydrophobicity. However, the reactive acyl chloride and anhydrides or vinyl esters are obtained from non-renewable sources, and previous investigations have required the utilization of anhydrous conditions and non-eco-friendly (even toxic) organic solvents, making the reaction difficult and time-consuming due to the difficulty of reducing the high water content of the reaction medium. Furthermore, these acylants are too expensive for use in large-scale industrial applications. In addition, the concurrent release of hydrohalic or carboxylic acid by-products from acyl chlorides and anhydrides can adversely impact the cellulose structure and/or contaminate the fiber surface [104,120]. As an efficient and environmentally friendly approach, plant-derived vinyl esters (plant oils) have been used in place of petroleum compounds. This strategy has attracted substantial attention in recent years because the chemicals are easier to handle, less toxic, noncorrosive, and produce less waste throughout the reaction [92].

Plant oils as transesterification agents are triglyceride fatty acid esters derived from abundant and renewable resources. These compounds have been mainly used in food applications as more sustainable reagents for synthesizing CELs. The first study utilizing plant oils directly for cellulose modification was by Dankovich et al., who investigated increasing the water repellency of cellulosic materials such as CMC powder, cellulose filter paper, and cotton fabrics. The authors modified the cellulosic materials with several plant oils (soybean, rapeseed, olive, coconut) in acetone, ethanol, or DI water emulsion combined with a surfactant at 110 °C for 1 h. All of the oils, except for coconut, produced a hydrophobic cellulose surface with less water absorption capacity. The most hydrophobic surfaces were obtained with 1% soybean oil in acetone. The WCA was improved to 81° and the WA was decreased to 0.82 mL/mg. The maximum decomposition temperature of the cellulose esters was increased by 21 °C compared with the original CMC [122]. Dong et al. carried out transesterification between CMC and soybean oil. The CMC and soybean oil were first dispersed in ethanol and sonicated for 1 min. The samples were then dried at RT and placed in an oven at 110 °C for 15–120 min. After the modification, the CMC displayed a higher affinity for low-polar solvents; additionally, the hydrophobicity of the cellulose esters could be adjusted by controlling the heating time [123]. In another study, CMC was uniformly distributed in an emulsion mixture of rice bran oil and ethanol by sonicating for 15 min, followed by RT drying and reacting in an oven at 110 °C for 30, 60, or 90 min without the addition of a catalyst. The percentage of C18-CEL acylation was 15.4% at 3.3 wt% rice bran oil and 90 min of treatment time. The WCA of C18-CEL was enhanced to 92°, and WA was reduced to 0.9 µL/mg after modification compared with 2.5 µL/mg for unmodified CMC. As the treatment time increased, the T_d-onset_ of C18-CEL decreased to 325, 311, and 306 °C, respectively, for 30, 60, and 90 min of treatment time, while the T_d-onset_ of the original CMC was 306 °C [124]. Onwukamike et al. carried out transesterification between high-oleic sunflower oil and CMC (*M*_W_ = 89.0 kDa), cellulose filter paper (FP, *M*_W_ = 190.0 kDa), and cellulose pulp (*M*_W_ = 132.0 kDa) in a DBU-DMSO-CO_2_ switchable solvent system at 115 °C for 6–24 h with DBU as a catalyst. The DS was increased from 0.34 to 1.59, as the reaction time was prolonged from 6 to 24 h. Among the investigated cellulose samples, the C18-CEL film from cellulose pulp (C18-CEL pulp) showed the highest E (478 MPa), followed by the C18-CEL FP film (458 MPa) and C18-CEL CMC film (376 MPa). Higher E values were obtained for films with lower DS values because the increasing presence of aliphatic side groups from the fatty acids increased the flexibility. All films showed an increased T_d-onset_ after modification. C18-CEL CMC had a T_d-onset_ of 320 °C (DS = 1.59) (compared with 280 °C for pristine CMC). C18-CEL FP had a T_d-onset_ of 329 °C (DS = 1.48) (compared with 298 °C for pristine FP). C18-CEL pulp had a T_d-onset_ of 327 °C (DS = 1.40) (compared with 261 °C for pristine pulp) [92].

The abovementioned transesterification processes required the use of organic solvents. The development of a greener process in water could therefore be beneficial from a sustainability point of view and be advantageous for industrial applications. Yoo et al. adopted a green procedure in aqueous lactic acid syrup for the acylation of CNC with plant oils (tung oil, linseed oil) with zinc acetate dihydrate and dibutyltin dilaurate as catalysts. The reactive solvent and intermediate product (CNC-*g*-PLA) system allowed for an in situ solvent exchange from DI water to lactic acid without prior drying of the CNC and facilitated the subsequent efficient acylation of CNC with plant oils. Approximately one third of the available -OHs on the CNC surface were substituted with PLA oligomers and plant oil. Furthermore, the side products derived from lactic acid can be recycled and reused, providing a simple, ecofriendly, and industrially amenable strategy for the hydrophobic modification of cellulose [80].

## 5. Crosslinking

Crosslinking can enhance the water stability, thermal stability, and mechanical properties of cellulose. In this strategy, crosslinked covalent linkages are formed between crosslinking agents and -OHs or functional groups on cellulose and cellulose derivatives (Table 7) [70].

### 5.1. Crosslinking on Hydroxyl Groups of Cellulose

In one study, the chemical crosslinking of softwood cellulose papers was achieved through dipping papers in aqueous solutions containing the polyfunctional carboxylic acids 1,2,3,4-butanetetra-carboxylic acid (BTCA), tricarballylic acid (TCA), or succinic acid (SA) using NaH_2_PO_4_ as a catalyst for 30 min. These treated cellulose papers were then cured in an oven at 150 °C for 5–30 min. BTCA showed better crosslinking capability than TCA. Compared with pristine paper, the wet σ of the cellulose papers increased with an increasing number of crosslinking sites. However, papers treated with SA showed very little wet strength enhancement because cellulose modification by SA resulted in the attachment of cellulosic units with a single pendant carboxylic group, which has low reactivity with the cellulosic -Ohs, and thus yields little crosslinking [129]. Citric acid (CAC) is non-toxic, and is widely used in the food industry as a safe natural additive. He et al. used CAC as a crosslinker to increase the hydrophobicity of CNC. An aqueous colloidal suspension of CNC and CAC was reacted in an oven at 130 °C for 4 h with NaH_2_PO_4_ as a catalyst. The WCA of CAC crosslinked CNC (CAC-CNC) was increased to only 55° (compared with 41° for CNC). However, the CAC-CNC had a higher thermal stability (T_d-onset_ = 298 °C) than unmodified CNC (T_d-onset_ = 249°) [130]. In another study, sorbitol plasticized nanocellulose-coated filter paper was crosslinked in a CAC aqueous solution and then cured in an oven at 150 °C for 5 min, using NaH_2_PO_4_ as a catalyst. When three layers of CAC crosslinked sorbitol-NC coating were applied, the resulting plasticized, crosslinked and coated filter paper exhibited a decreased WVP and OP of 0.5 g mm kPa^−1^ m^−2^ day^−1^ and 2 mL µm m^−2^ day^−1^ kPa^−1^, compared with 4.0 g mm kPa^−1^ m^−2^ day^−1^ and 197 mL µm m^−2^ day^−1^ kPa^−1^ for unmodified filter paper, respectively. The modified paper did not show obvious changes in σ, but its E decreased from 570 to 310 MPa and the ε slightly increased from 2.5% to 3.7% compared with the control paper. The T_d-onset_ of the modified paper decreased from 311 °C to 288 °C in comparison with that of the non-modified paper [125].

### 5.2. Crosslinking with Functional Groups of Cellulose Derivatives

Cellulose derivatives bear various functional groups (e.g., hydroxy, propyl, methyl, carboxymethyl, and formyl groups). When reacted at 190 °C for 15 min with PEG400 as a plasticizer and NaH_2_PO_4_ as a catalyst, 5 wt% CAC as a crosslinker for hydroxy propyl methyl cellulose (HPMC) decreased the WS of CAC crosslinked HPMC (CAC-HPMC) films by 74% compared with pristine HPMC film, and 15 wt% CAC resulted in the highest reduction in WVTR by 47% for CAC-HPMC film compared with neat HPMC film [126]. Another study modified carboxymethyl cellulose (CMCS) in two ways. One method used photo-crosslinking via UV irradiation at RT for 30–180 min with sodium benzoate (SB) as a photo-initiator. The other method involved chemical crosslinking using saturated glutaraldehyde (GLA) vapor plus gelatin as a synergistic crosslinker at 80 °C for 30–180 min. The photo-crosslinked film treated with 20 wt% SB with an irradiation time of 180 min and the chemically crosslinked film modified by adding 0.2 g of gelatin and exposed to GLA vapor for 90 min were identified as having the best crosslinking degrees. The photo-crosslinking treatment was more effective than the chemical crosslinking process in terms of WCA, WVP, and σ. WS of the CMCS/SB/UV and CMCS/GLA/gelatin films were reduced to 57.1% and 50.1%, respectively, compared with 78.2% for the pristine CMCS film. The CMCS/SB/UV film had a slightly higher WCA (78°) than the CMCS/GLA/gelatin film (67°). WVP of the CMCS/SB/UV film was greatly decreased to 9 × 10^−7^ g m^−1^ s^−1^ Pa^−1^ (compared with 819 × 10^−7^ g m^−1^ s^−1^ Pa^−1^ for the neat CMCS film), while the CMCS/GLA/gelatin film showed a smaller decrease in WVP (to 50 × 10^−7^ g m^−1^ s^−1^ Pa^−1^). σ of the CMCS/SB/UV and CMCS/GLA/gelatin films was considerably enhanced to 46.8 MPa and 33.7 MPa, respectively, compared with 14.2 MPa for the unmodified CMCS film. However, ε of the CMCS films decreased from 19.9% to 9.1% and 13.8% after crosslinked with SB/UV and GLA/gelatin, respectively. The photo-crosslinked and chemically-crosslinked films showed no cytotoxicity [127]. In another study, softwood CNF was periodate-oxidized to dialdehyde CNF (DACNF) by sodium periodate with 6% of AGU on cellulose oxidized. Through the Maillard reaction, gelatin was crosslinked to DACNF at 60 °C for 3 h. The gelatin-crosslinked DACNF (G-DACNF) film (crosslinking degree = 57%) showed markedly decreased WA when immersed in DI water for 1 h, resulting in a weight increase of only 44%, while the neat CNF film weight increased by 167%. The G-DACNF film also exhibited a much higher wet mechanical strength (wet ε: 23.2%, wet σ: 15.4 MPa) than unmodified CNF film (6.7%, 0.9 MPa) and DACNF (16.5%, 9.9 MPa) [57]. In another study, chitosan was used as the crosslinker for DACNF. There was a slight increase in the WVP of the crosslinked films compared with that of the raw cellulose film. The OP of chitosan crosslinked films was greatly reduced to 4.3–8.9 × 10^−15^ cm^3^ cm/cm^2^ s Pa, while the OP of the pure cellulose film was beyond the detection limit of the OP analyzer. In addition, σ of the crosslinked films was much higher than that of the unmodified film [131]. Furthermore, 4-(4,6-Dimethoxy-1,3,5-triazin-2-yl)-4-methyl-morpholinium chloride (DMTMM) was found to be an efficient crosslinking agent that promoted the reaction between carboxylic groups and -OHs under mild reaction conditions to form esters. With gelatin as a plasticizer, Beghetto et al. employed DMTMM as a crosslinking agent for CMCS. The reaction was carried out in DI water at RT for 2 h, resulting in a transparent DMTMM crosslinked CMCS film (DMTMM-CMCS) with an optical transmittance of 80–90%. By-products formed from DMTMM during the reaction were 2,4-dimethoxy-6-hydroxy-1,3,5-triazine (DMTOH) and *N*-methyl morpholinium hydrochloride, which are nontoxic and can easily be removed by water washing. A further advantage is that DMTOH can be recovered and recycled to regenerate DMTMM. Furthermore, 5 wt% DMTMM-CMCS film showed the lowest WS (18.1%), while CMCS completely dissolved in DI water within 4 h. The 10 wt% DMTMM-CMCS film had a decreased WVP and oil resistance ability (0.68 × 10^−7^ g m^−1^ h^−1^ Pa^−1^ and 0.29%), which was considerably lower than that of the unmodified CMCS film (1.14 × 10^−7^ g m^−1^ h^−1^ Pa^−1^, 1.89%). Increasing amounts of DMTMM led to significantly improved σ, although the ε gradually decreased. With the addition of 5 wt% DMTMM, the σ was enhanced from 32 MPa to 54.9 MPa, while the ε decreased from 30.1% to 25.4% compared with the neat CMCS film. Adding 50 wt% glycerol as a plasticizer to the 5% DMTMM-CMCS film achieved optimum σ and ε (52.3 MPa, 37.3%), satisfying the need for a high σ and a high ε for food packaging applications. The crosslinked CMCS film also showed higher thermal stability than the unmodified CMCS film [128]. Crosslinking methods of cellulose derivatives, reaction degree, and the associated packaging forming methods are listed in Table 8. And the crosslinked cellulose materials used in food packaging applications are listed in Table 9.

## 6. Degradability of Esterified, Cross-Linked Cellulose and Its Derivatives

Products featuring cellulosic materials often advertise their complete biodegradability, which is in contrast with traditional petroleum-based plastics. According to the 94/62 EC Directive, a material can be defined as biodegradable if 90% of the material decomposes naturally within 6 months when used as compost [132]. The enzymatic biodegradation of cellulose has been reported to proceed completely within 60 d, during which the microorganisms break down the cellulose backbones and transform the glucose units into CO_2_, water, and CH_4_ [133]. Increasing the production of cellulose packaging to meet demand requires the hydrophobic modification of cellulose to improve the structural stability of the cellulose material in high humidity environments. The chemical modification of cellulose through (trans)esterification or crosslinking has been proven feasible.

The literature shows that the key architectural properties for biodegradability are the molecular weight (*M*_w_) and crystallinity. In general, there is an inverse relationship between the mechanical performance and the biodegradability of cellulose and petroleum-based plastic packaging. Increasing the *M*_w_ and crystallinity is generally associated with better mechanical performance but decreased biodegradability, while lower crystallinity corresponds to looser chain packing, facilitating enzyme access [116]. For covalently modified cellulose, the biodegradability depends on both the DS and the nature of the chemical linkages (e.g., ether, ester). Studies have found that cellulose functionalized with a range of esters, ethers, or crosslinked moieties exhibits decreased rates and extents of biodegradation compared with unmodified cellulose, with etherified cellulose exhibiting the highest recalcitrance. One study reported that CNF ethers became nonbiodegradable at low surface DS values (≈0.1), while the biodegradability of CNF esters at similar DS values was less affected [133]. The removal of functional groups (e.g., through the hydrolysis of ester groups) before attacking the -OHs present in native cellulose is the major rate-limiting step in the biodegradation of functionalized cellulose.

While native cellulose is readily and fully biodegraded, esterification has the potential to interfere with its biodegradability. Bare CA film partially degraded in soil with a 65% weight loss after 58 d, while pristine cellulose film completely degraded in soil within 60 d. When CA films were incorporated with additives such as sodium alginate or carrageenan, the composite films degraded at a slower rate. CA-sodium alginate and CA-carrageenan films showed 55% and 50% weight loss after 58 d, respectively [134]. One study found that the moderately substituted hexyl-esterified CNF (DS = 1.19) exhibited comparable biodegradation to unmodified CNF, while a highly functionalized C6-CES sample (DS = 2.43) showed 70% biodegradation compared with unmodified CNF. Dodecyl-esterified CNF (DS = 2.46) displayed considerably lower biodegradation of only 37% compared with unmodified CNF, indicating that cellulose esters with similar DS values but increasing substituent chain lengths biodegrade to a much lesser extent [133]. Fully substituted cellulose esters (C12–C20) were found to display little biodegradability [87].

The use of crosslinking to reinforce cellulose is thought to also significantly reduce the overall biodegradability, but there are few studies on this topic. One study found that DMTMM as a crosslinker contributed to a more compact and resistant structure, thereby decreasing the biodegradability rate of cellulose-based films [128]. The films crosslinked with 5 wt% DMTMM only began to biodegrade on day 7 and did not completely degrade in soil until day 15, while neat CMCS films degraded entirely by day 7. However, the addition of highly hydrophilic glycerol to cellulose film increased its water retention and consequently accelerated its biodegradability.

## 7. Conclusions

The overuse of petroleum-based plastics in the past few decades has led to growing concerns about environmental pollution. In recent years, research and innovation on cellulose-based food packaging have provided solutions to help reduce our dependency on fossil fuel-based packaging films. The advantages of cellulose films are the abundant natural origin of cellulose and its complete biodegradability. However, the ability of chemical modification to mitigate the hydrophilic nature of cellulose and to ensure that the mechanical and barrier performance of cellulose-based packaging is comparable to that of petroleum-based counterparts, while maintaining the biodegradability of the films, must be exploited and evaluated. Several mechanisms have been used to achieve mechanical and barrier performances that are equivalent to, and even exceed those of, corresponding petroleum-based plastics such as LDPE. Methods for producing active cellulose packaging (e.g., antioxidant, antimicrobial packaging) have also been developed. Blending cellulosic material with additives such as plasticizers and active agents has been demonstrated to improve the mechanical features and prolong the food storage ability of cellulose-based packaging without greatly reducing its biodegradability.

With respect to industrial-scale applicability, chemically modified hydrophobic cellulose packaging has not yet been adequately explored to establish its capacities for current and future applications. Given the range and diversity of options available for chemical hydrophobization strategies, there are excellent prospects for extending the application range of cellulose-based plastics on a scale comparable to that of fossil fuel-based thermoplastics and beyond. Highly hydrophobic cellulosic packaging, without the drawbacks of resource depletion, pollution, and recalcitrance, has the potential to act as a potent alternative in the food packaging market, and could contribute to future prosperity for the planet and its inhabitants.

## Figures and Tables

**Figure 1 polymers-14-01533-f001:**
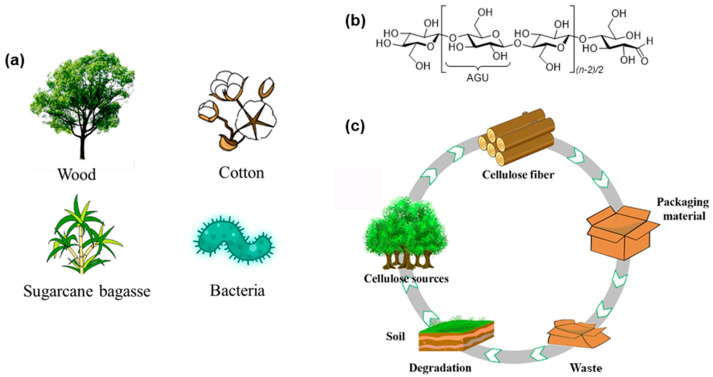
(**a**) Common sources of cellulose; (**b**) Chemical structure of cellulose; (**c**) Life cycle of cellulosic food packaging [15].

**Figure 2 polymers-14-01533-f002:**
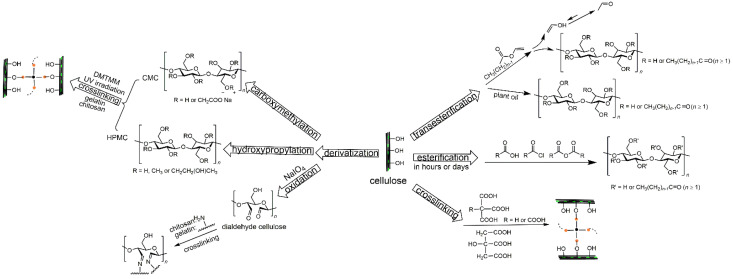
(Trans)esterification and crosslinking reaction schemes of cellulose and cellulose derivatives.

**Figure 3 polymers-14-01533-f003:**
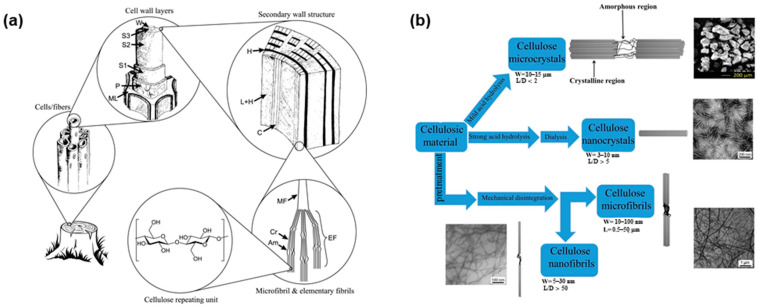
(**a**) Hierarchical structure of wood, showing: the middle lamella (ML), the primary wall (P), the outer (S1), middle (S2), and inner (S3) layers of secondary wall, the warty layer (W), cellulose (C), hemicellulose (H), lignin (L), microfibril (MF), elementary fibril (EF), crystalline domain (Cr) and amorphous domain (Am) [65]; (**b**) Methods to manufacture cellulose fibers with various sizes [17].

**Figure 4 polymers-14-01533-f004:**
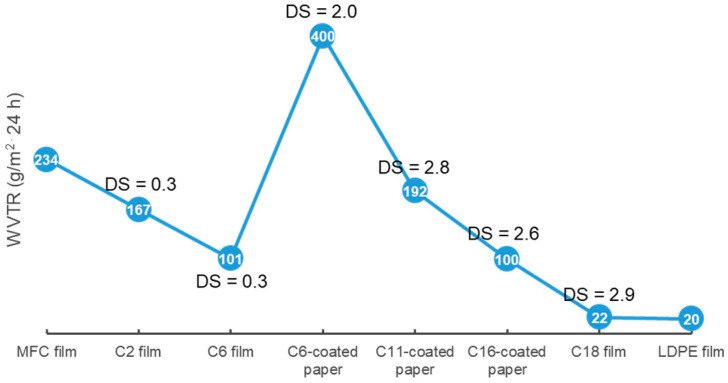
WVTR of CMF film, cellulose ester packaging and LPDE film [5,96,102,105,109].

**Figure 5 polymers-14-01533-f005:**
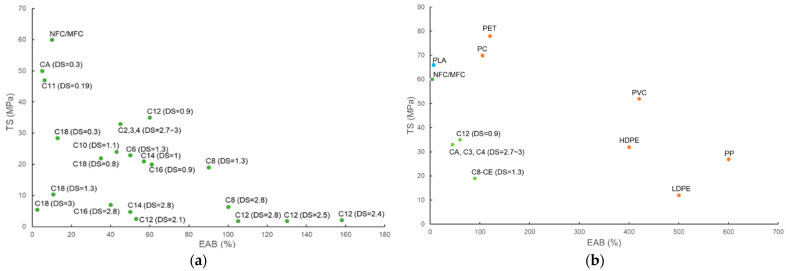
(**a**) Mechanical strength (ε *vs.* σ) of cellulose and cellulose esters with various chain lengths and DS values; (**b**) mechanical strength (ε *vs.* σ) of cellulose and cellulose esters compared with poly(lactic acid) (PLA) and petroleum-derived plastics [116].

**Figure 6 polymers-14-01533-f006:**
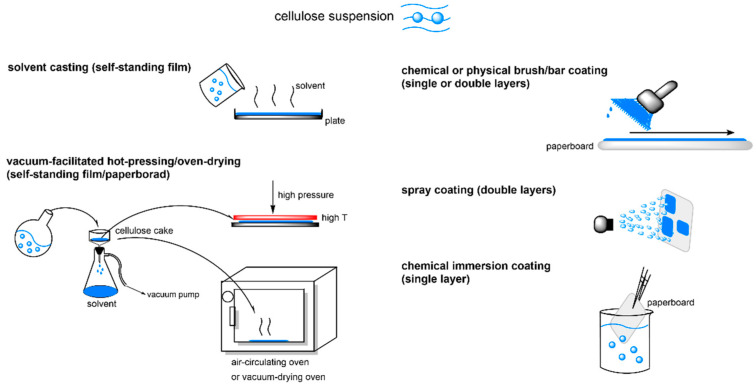
Strategies for cellulose ester-based packaging manufacture.

**Table 1 polymers-14-01533-t001:** Extraction methods and properties of cellulose from diverse natural sources.

Cellulose Source	Extraction Method	Fiber Size	Crystallinity	Functionality	Reference
kenaf bast fiber	disintegration, refining, cryo-crushing and cylinder homogenization	CNF, diameter: 10–30 nm	81%	–OH	[76]
*Gluconacetobacter xylinus* NRRL B-42	blender homogenization	BNF, fiber size: n.d. ^a^	91.8%	–OH	[77]
tunicin cellulose	TEMPO ^b^-mediated oxidation, blender homogenization, sonication	CMF, diameter: 10–20 nm	n.d.	–OH and sodium carboxylate groups (0.31 mmol/g)	[78]
wood pulp filter paper	blender homogenization, refining, freeze-drying	CNC, fiber size: n.d.	85%	–OH and sulfate groups	[79]
commercial never-dried CNC suspension in water	-	CNC, length: 64 nm, width: 7 nm	72%	–OH and sulfate groups	[80]
wheat straw CMF	blender homogenization	CNF, diameter: 10–40 nm	89%	–OH	[62]
softwood pulp dissolved with sulfite	carboxymethylation and cylinder homogenization	CMF, diameter: 5–15 nm	n.d.	–OH and sodium carboxylate groups (586 μ-equiv./g)	[81]
softwood and hardwood bleached kraft pulp	TEMPO-mediated oxidation, blender homogenization, sonication	TOCN ^c^, length: several μm, diameter: 3–4 nm	75%	-OH and sodium carboxylate groups	[82]
spruce/pine (*w*/*w* = 7/3) bleached softwood pulp	enzymatic treatment and cylinder homogenization	CNF, diameter: 20 nm	n.d.	-OH and sulfate groups	[83]
CMC from cotton linters	CNC: acid hydrolysis using sulfuric acid and freeze-dried, regenerated cellulose: treated by *N*-Methylmorpholine *N*-oxide under heating	CNC (length: 300 nm, diameter: 10 nm), regenerated cellulose (length: 100 μm)	80%	CNC: –OH and sulfate groups (70 mmol kg^−1^)	[84]

^a^ Not detected; ^b^ 2,2,6,6-Tetramethyl-1-piperidinyloxy; ^c^ TEMPO-oxidized CNF.

**Table 2 polymers-14-01533-t002:** Esterification of cellulose using C2–C6 acylants, reaction degree, and the associated packaging forming methods.

Cellulose	Acylation Process	Packaging Type and Its Formation	Reaction Degree	Ref.
softwood cellulose pulp; CMF	acetic acid or acetic anhydride as acylant with or without sulfuric acid as catalyst, reacted at 60–70 °C for 0.5–4 h with or without toluene as solvent	CA coating or film; CA solution was coated on paper via the hand lay-up technique or solvent-casted in air	DS = 0.21–0.32 for C2-CMF CES	[93,97]
CMF	oxalic acid as esterifying agent, reacted at 90 °C for 0.25–4 h in DI water	C2-CES coating; C2-CES aqueous suspension was deposited on filter paper through vacuum filtration, then oven-dried	carboxyl group content of 0.21–0.43 mmol/g fibrils	[89]
CNC extracted from bamboo waste pulp	butyric anhydride as acylant and iodine as catalyst, reacted at 105–110 °C for 30 min without solvent	C4-CES film or coating; C4-CES solution was solvent-casted or coated on mung bean seeds	DS = 2.1	[94]
cotton cellulose	acetic, propionic or butyric anhydride as acylant with trifluoroacetic acid as solvent/catalyst, reacted at 50 °C for 1 h	C2, C3, C4 (or their mixture)-CES films or coatings; CES solution was solvent-casted or coated on white paper with a bar coater	DS = 2.7–3	[95]
CMF from wood pulp	hexanoyl chloride as acylant and activated by mechanical ball milling, reacted at RT for 1–12 h in DMF	C6-CES film; C6-CES solution was solvent-casted, dried in an air-circulating oven at 60 °C and treated at 60 °C under vacuum for another 8 h	DS = 0.25–0.45 (3 h of milling time, 1–4 mL of acylant)	[96]

**Table 3 polymers-14-01533-t003:** Overview of C2–C6 CES materials in food packaging applications.

Cellulose Packaging	Additives	Packaging Properties			Food Packaging Application	Ref.
		Barrier Properties	Mechanical Properties	Other Properties		
kraft paper (CA as coating)	2–8 mL of CIN (*v*/*v* in CA solution)	oil resistance: kit number 12; WA, WVP and OTR of the CIN-CA-coated paper markedly decreased by 96.2%, 76.8%, and million times, respectively	dry and wet σ was increased from 55.8 and 2.3 MPa to 88.2 and 12.9 MPa, respectively, compared with kraft paper	good cytocompatibility, high antioxidation with 8% CIN, excellent antibacterial performance with 6% CIN	extend beef’s shelf-life by 4–5 d at 4 °C	[97]
CA-coupled cellulose cardboard (food-grade resin as coating)	10% (*w*/*w* to resin) LDH-HB	n.d. ^a^	n.d.	good cytocompatibility in the release test, excellent antibacterial performance	preserved cooked pasta for up to 30 days at 4 °C	[5]
CA film	10–50 wt% PEG, 1–5 wt% MMTCTAB-MMT or 0.05 wt% G- AgNPs	3 wt% CTAB-MMT-incorporated CA film showed the lowest WVTR of 5.84 g/m^2^ 24 h; 0.05% G-AgNP-incorporated CA film showed an increased degree of water swelling from 0.28 to 0.44–0.62	ε of 20 wt% PEG-incorporated CA film (CAP20) was increased from 3.8% to 31.0%, while σ decreased from 43.3 MPa to 32.6 MPa compared with CA film	CAP20 film incorporated with CTAB-MMT showed slightly increased thermal stability, good antimicrobial properties, and no cytotoxicity; G-AgNP-incorporated CA film showed strong antibacterial activity and no cytotoxicity	n.d.	[37,98]
CA film	0.1–1% of carotenoids (lycopene, norbixin and zeaxanthin) (*w*/*w* to CA)	0.1 wt% carotenoids: WVP = 0.035 g × mm m^−2^ h^−1^ kPa^−1^ for norbixin, 0.023 g × m m^−2^ h^−1^ kPa^−1^ for zeaxanthin and 0.022 g × mm m^−2^ h^−1^ kPa^−1^ for lycopene	0.1 wt% lycopene or zeaxanthin: ε increased from 3.9% to 15%, while ε of 0.1 wt% norbixin-CA film remained unchanged; σ increased from 65.3 MPa to 84, 104 and 86 MPa, respectively, for norbixin-, lycopene- and zeaxanthin-loaded films	films with norbixin or lycopene displayed better light protection for sunflower oil; films with norbixin showed the best UV protection for vitamin B_2_; film with 0.1% zeaxanthin showed 50 °C lower T_d-onset_ (200 °C) compared with CA film and the films with other carotenoids	n.d.	[99,100]

^a^ Not detected.

**Table 4 polymers-14-01533-t004:** Esterification of cellulose using C2–C20 acylants, reaction degree, and the associated packaging forming methods.

Cellulose	Acylation Process	Packaging Type and Its Formation	Reaction Degree	Ref.
unbleached eucalyptus CNF; wheat bran and maize bran residue cellulose; cellulose sheet; softwood cellulose; CMC; α-cellulose	C6–C20 acyl chlorides as acylants, pyridine, sulfuric acid or DMAP as catalyst, reacted at 50–130 °C in DMAc, DMAc/LiCl or cosolvent of toluene and pyridine for hours	C6–C20 films; solvent casting or vacuum dried	DS = 0.19–3 (DS of cellulose esters decreased with increasing substituent chain length from C6 to C18)	[87,102,104,106,107,112,113,115]
BNF, α-cellulose	C2–C12 carboxylic acids or C8–C18 acyl chlorides as acylants, reacted at 50–130 °C for 2 h in pyridine or pyridine/tosyl chloride	C2–C12 cellulose ester papers or C8–C18 cellulose ester films; wet cellulose cakes or films were hot-pressed at 90–110 °C	DS = 0.64–3 (DS of cellulose esters decreased with increasing substituent chain length from C2 to C12)	[103,107]
*α*-cellulose, bleached bagasse pulp, CMC	C6–C16 acyl chlorides as acylants and pyridine as catalyst, reacted in DMAc, DMAc/LiCl or pyridine at RT–100 °C for hours or days	C6–C16 cellulose ester-coated paper; cellulose ester suspension or solution was spray- or bar-coated on paperboard via air brush or bar coater	DS = 0.62–2.9	[105,108,109]
premade CNF film	lauroyl, palmitoyl or stearoyl chloride as acylant, reacted in pyridine at 100 °C	C12-, C16- and C18-cellulose ester films; one-sided acylation using a brush and reacted at 100 °C; immersion acylation at 100 °C for 90 min	immersion modification resulted in a higher DS (0.91–1.8) than one-sided modification (0.37–0.55)	[110]
oat straw CNF	acetic, butyric, hexanoic or 2-dodecen-1-yl-succinnic anhydride as acylant, reacted at 80 °C in an oven for 2 h with a 10 kg weight on top	C2–C16 cellulose ester nanopapers; CNF wet cake was immersed in acylant liquid for 2 d, then put in an oven at 80 °C for 2 h under hot pressing	DS of cellulose esters decreased from 0.38 to 0.1 with increasing substituent chain length from C2 to C16	[114]

**Table 5 polymers-14-01533-t005:** Overview of cellulose ester (C2–C18) materials in food packaging applications.

Cellulose Packaging	Packaging Properties			Ref.
	Barrier Properties	Mechanical Properties	Other Properties	
C11-CEM film (DS = 0.19)	higher WCA (101°) and lower WA (6%) than original CNF film (54°, 95%), lower WVP at 3.4 × 10^−9^ g⋅m^−1^ s^−1^ Pa^−1^ than pristine CNF film (9.0 × 10^−9^ g⋅m^−1^ s^−1^ Pa^−1^)	decreased σ and E (47 MPa, 2075 MPa) compared with that of neat CNF film (57, 3847 MPa), while the ε was slightly increased (6.2% *vs.* 4.5%)	T_d-onset_ was increased slightly to 350 °C compared with that of CNF (343 °C)	[115]
C18-CEL film (DS = 2.53–2.86)	WVTRs of isostearic-, oleic- and modified TOFA- cellulose ester films were markedly reduced to 21.7, 22.4 and 43.4 g/m^2^ 24 h, poor oxygen resistance (too high for the sensor)	isostearic CEL film showed the highest ε of 101% (twice that of the original cellulose film), while oleic CEL film and modified TOFA CEL film had lower εs (57% and 45%)	T_d-onset_ of C18-CELs (327~340 °C) was 7~22 °C higher than that of unmodified cellulose (320 °C)	[102]
C18-CEL film (DS = 0.3–3)	C18-CEL (DS = 0.3) film absorbed less water (13.9%) than pristine CMC film (28.9%), while C18-CEL (DS = 3) did not show significant WA, C18-CEL3 film had higher WCA (110°) than C18-CEL0.3 film (102°), WVPs of C18-CEL3 and C18-CEL0.3 films were decreased to 5 × 10^−12^ and 75 × 10^−12^ g m^−1^ s^−1^ Pa^−1^, respectively, compared with 158 × 10^−12^ g m^−1^ s^−1^ Pa^−1^ for neat CMC film	C18-CEL0.3 film had higher mechanical strength (σ = 28.5 MPa, E = 1118 MPa, ε = 12.7%) than C18-CEL3 film (σ = 5.5 MPa, E = 286 MPa, ε = 2.5%), but both substituted films had decreased mechanical strength compared with unmodified CMC film (σ = 169.2 MPa, E = 7230 MPa, ε = 15.9%)	C18-CEL0.3 film exhibited moisture-responsiveness, while C18-CEL3 film showed thermal responsiveness, the C18-CEL3 film displayed a distinct T_m_ at 55 °C, while no T_m_ was found for the C18-CEL0.3 film, reversible changes in the C18-CEL3 film volumes were observed when varying the temperature	[104]
C6-, C8-, C10-, C12-, C14- and C18-cellulose ester films (DS = 0.8–1.3)	WCA was increased from 66° to 90°, WVP was decreased from 6 to 1.6 cc × mm m^−2^ d^−1^ kPa^−1^ for C6 to C18, which was markedly lower than 20–25 cc × mm m^−2^ d^−1^ kPa^−1^ for pristine CNF film, C6–C18 cellulose ester films all exhibited poor oxygen resistance	C6-CES, C10-CEM and C12-CEM films had higher E (550–600 MPa) compared with other cellulose ester films, C8-CEM (DS = 1.3) film had the highest ε (90%), C12-CEM (DS = 0.9) film had the highest σ (35 MPa); the other cellulose ester films had lower σ (20–24 MPa)	cellulose ester films were transparent, flexible and heat-sealable, melted at 170–225 °C, and were able to be squeezed through a 2-mm rod die	[106]
C11-CEM-coated paper (DS = 2.75)	WCA decreased from 117° to 101° with coating grammage increasing from 0.97 to 6.25 g m^−2^, WVTR decreased from 441 to 192 g/m^2^ 24 h with increasing coating grammage, compared with 622 g/m^2^ 24 h of uncoated paper	slightly increased ε an nonsignificant change in tensile index	PHGH and MPA were attached to C11-CEM-coated paper, giving the paper desirable antimicrobial performance	[109]
C6- and C16-cellulose ester-coated paper (DS = 1.6–2.9)	WCA was enhanced to 95–123°, WVTR of C16-CEL-coated paper (100–300 g/m^2^ 24 h) was considerably lower than that of C6-CES-coated paper (400–1020 g/m^2^ 24 h)	cellulose ester-coated paper showed slightly enhanced σ (12.4–12.7 MPa) compared with uncoated paper (11.8 MPa)	C16-CEL-coated paper showed higher T_d-onset_ (350 °C) than C6-CES-coated paper (320 °C). C6-CES and C16-CEL powder became liquid at 160 °C and 220 °C, respectively, in an oven	[105]
C12-, C16- and C18-cellulose ester films, DS (immersion method) = 0.9–1.8, DS (one-sided method) = 0.37–0.55	WCA of one-sided acylated film: 105–121°, WCA of immersion-acylated film: 113°, WVP of one-sided acylated film did not show any decrease compared with that of unmodified CNF films (WVP = 0.057 ng s^−1^ m^−1^ Pa^−1^), WVP of immersion-acylated film decreased to 0.006–0.021 ng s^−1^ m^−1^ Pa^−1^	mechanical strength of one-side-acylated film was comparable to that of the original CNF film, immersion-acylated film showed much weaker mechanical strength	thermal stability was increased for immersion-acylated films, whereas that of one-side-acylated films was similar to pristine CNF film (T_d-onset_ = 245 °C)	[110]
C2-, C4-, C6- and C16-cellulose ester nanopapers (DS = 0.1–0.38)	WCA was enhanced from 32° to 119° from C2 to C16 and was 24° for the neat CNF nanopaper, C16-CEL paper floated on the water surface for several weeks, while pristine CNF paper sank	C2-CES nanopaper had the highest E, and C16-CEL nanopaper had the lowest E, C16-CEL had the highest wet strength, which was 7- fold greater than that of the reference CNF nanopaper	n.d. ^a^	[114]

^a^ Not detected.

**Table 6 polymers-14-01533-t006:** Influence of packaging manufacturing technologies on properties of cellulose ester-based packaging (DS = 0.2–2.8).

Technology	Smoothness	WCA ^a^	WVP	TS	Efficiency	Ref.
self-standing film	‘4–5’ ^b^	84–107°, ‘2–3’	reduced 62–90%, ‘4–5’	decreased 18%, ‘4’	‘2’	[103,106,115]
physical bar coating on paperboard	‘3–4’	100–124°, ‘3–4’	reduced 29~69%, ‘2–4’	no change, ‘5’	‘1’	[105,109]
chemical brush coating on film	‘4’	109°, ‘3’	no change, ‘1’	decreased 32%, ‘3’	‘4’	[110]
spray coating on paperboard	‘1’	152°, ‘5’	no change, ‘1’	n.d. ^c^	‘3’	[108,109]
chemical immersion coating on film	‘3’	114°, ‘3’	reduced 63~90%, ‘4–5’	decreased 68~80%, ‘1–2’	‘5’	[110]

^a^ WCA of neat cellulose film or paperboard is 15–57°; ^b^ With increasing number, cellulose ester-based packaging displays poorest (‘1’) or best (‘5’) performance for each parameter, and ‘2–4’ is the extent between them; ^c^ Not detected.

**Table 7 polymers-14-01533-t007:** Influence of crosslinking on water sensitivity and mechanical strength of cellulose packaging (after vs. before).

	Parameters	Crosslinker	Water Solubility (WS)/WA	WVTR/WVP	σ, ε	Ref.
Cellulose						
sorbitol plasticized nanocellulose-coated filter paper		CAC	n.d. ^a^	WVP: reduced by 88%	no change in σ; ε: 3.7% *vs.* 2.5%	[125]
HPMC		CAC	WS: reduced by 74%	WVP: reduced by 43%	n.d. ^a^	[126]
CMCS		UV irradiation	WS: reduced by 21%	WVTR: reduced by 99%	σ: 46.8 MPa *vs.* 14.2 MPa; ε: 9.1% *vs.* 19.9%	[127]
DACNF		gelatin	WA: 44% *vs.* 167%	WVP: reduced by 99.9%	wet σ: 15.4 MPa *vs.* 9.9 MPa; wet ε: 23.2% *vs.* 16.5%	[57]
CMCS		DMTMM	WS: 18% *vs.* 100%	WVP: reduced by 40%	σ: 55 MPa *vs.* 32 MPa; ε: 25% *vs.* 30%	[128]

^a^ Not detected.

**Table 8 polymers-14-01533-t008:** Crosslinking of cellulose derivatives, reaction degree, and the associated packaging forming methods.

Cellulose	Crosslinking Process	Packaging Type and Its Formation	Reaction Degree	Ref.
PEG400 plasticized HPMC	CAC as crosslinker and NaH_2_PO_4_ as catalyst, reacted in a mixture of DI water and ethanol with homogenization for 15 min, dried at 60 °C for 60 min and cured at 190 °C for 15 min	CAC-HPMC film; cured at high temperature	crosslinking rate ranged between 0% and 65% with a CAC content of 0–15% (*w*/*w* to HPMC)	[126]
CMCS	photo-crosslinking via UV irradiation at RT for 30–180 min with SB as photo-initiator or chemical crosslinking with saturated GLA vapor plus gelatin as synergistic crosslinkers at 80 °C for 30–180 min	CMCS/SB/UV or CMCS/GLA/gelatin film; films were prepared by the casting method at 45 °C for 18 h	photo-crosslinked film treated with 20 wt% SB and irradiated for 180 min, or chemically-crosslinked film modified with 0.2 g gelatin and exposed to GLA vapor for 90 min were found to have optimized crosslinking degrees	[127]
DACNF	gelatin as crosslinker, reacted in DI water at 60 °C for 3 h	G-DACNF film; vacuum filtration and solvent casting	crosslinking degree = 57%	[57]
gelatin plasticized CMCS	DMTMM as crosslinker for CMCS, reacted in DI water at RT for 2 h	DMTMM-CMCS film; films were prepared by the casting method at 40 °C overnight	optimum crosslinking degree was achieved in the presence of 5 wt% DMTMM and 50 wt% glycerol	[128]

**Table 9 polymers-14-01533-t009:** Overview of crosslinked cellulose materials in food packaging applications.

Cellulose Packaging	Additives	Packaging Properties			Ref.
		Barrier Properties	Mechanical Properties	Other Properties	
BTCA, TCA, SA crosslinked paper	-	n.d. ^a^	wet tensile index of BTCA crosslinked (27 mN/g) and TCA crosslinked paper (16.5 mN/g) were markedly enhanced compared with 1.3 mN/g of pristine paper, while papers treated with SA showed little wet strength enhancement	n.d.	[129]
CAC-CNC-coated filter paper (three layers of coating)	sorbitol	modified paper showed increased WA (37%) compared with uncoated filter paper (29%), WVP and OP of modified paper was considerably decreased to 0.5 g mm kPa^−1^ m^−2^ day^−1^ and 2 mL µm m^−2^ day^−1^ kPa^−1^ compared with 4 g mm kPa^−1^ m^−2^ day^−1^ and 197 mL µm m^−2^ day^−1^ kPa^−1^ of uncoated filter paper	modified paper had a nearly unchanged σ, a reduced E (from 570 to 310 MPa) and increased ε (from 2.5% to 3.7%) compared with the control filter paper	T_d-onset_ decreased from 311 °C to 288 °C	[125]
CAC-HPMC film	PEG400	increasing CAC content decreased WS of films, with an optimum CAC content of approximately 14%; 5 wt% CAC (*w*/*w* to HPMC) reduced WS of CAC-HPMC films by 74%; 15% CAC loading resulted in the highest reduction of WVTR by 47% for CAC-HPMC film (168 g/m^2^ 24 h) compared with neat HPMC film (316 g/m^2^ 24 h)	n.d.	CAC-HPMC films were transparent	[126]
CMCS/SB/UV or CMCS/GLA/gelatin film	-	WS of CMCS/SB/UV and CMCS/GLA/gelatin films was reduced to 57.1% and 50.1%, respectively, compared with 78.2% for pristine CMCS film, WVP of CMCS/SB/UV film was markedly decreased to 9 × 10^−7^ g m^−1^ s^−1^ Pa^−1^ compared with 8.19 × 10^−5^ g m^−1^ s^−1^ Pa^−1^ for neat CMCS film, while CMCS/GLA/gelatin showed a smaller WVP decrease (to 50 × 10^−7^ g m^−1^ s^−1^ Pa^−1^)	σ of CMCS/SB/UV and CMCS/GLA/gelatin films were considerably enhanced to 46.8 MPa and 33.7 MPa, respectively, compared with 14.2 MPa of unmodified CMCS film. However, the ε of the CMCS films decreased from 19.9% to 9.1% and 13.8% after crosslinking with SB/UV or GLA/gelatin, respectively	both crosslinked films were noncytotoxic	[127]
G-DACNF film	-	G-DACNF displayed greatly decreased WA when immersed in DI water for 1 h, resulting in a weight increase of only 44%, while the neat CNF film weight increased by 167%	G-DACNF film exhibited a much higher wet mechanical strength (ε: 23.2%, wet σ: 15.4 MPa, wet E: 94 MPa) compared with unmodified CNF film (6.7%, 0.9 MPa, 26 MPa)	G-DACNF films were transparent	[57]
DMTMM- CMCS film	gelatin (added or not)	5 wt% DMTMM-CMCS showed the lowest WS (18.1%); while CMCS completely dissolved in water within 4 h, the 10 wt% DMTMM-CMCS film showed a decreased WVP and oil absorption (0.68 × 10^−7^ g m^−1^ h^−1^ Pa^−1^ and 0.29%); however, the addition of glycerol increased the water sensitivity of the DMTMM-CMCS films	σ was enhanced from 32 MPa to 54.9 MPa, while the ε decreased from 30.1% to 25.4% with the addition of 5 wt% DMTMM compared with neat CMCS film. With the addition of glycerol, the optimum σ and ε values (52.3 MPa, 37.3%) were achieved in the presence of 5 wt% DMTMM and 50 wt% glycerol	DMTMM-CMCS films were transparent with optical transmittance of 80–90%	[128]

^a^ Not detected.

## Data Availability

All supporting data are reported in the manuscript.

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
