# Peer review of "Recent Advances in Chemically Modified Cellulose and Its Derivatives for Food Packaging Applications: A Review"

_polymers, 2022, doi:10.3390/polym14081533_

Round 1

Reviewer 1 Report

This work systematically summarized chemically modified cellulose and its derivatives for food packaging applications. The Reviews make a good contribution to the community, and the contents presented are of high quality. In my opinion, the manuscript can be accepted for publication after adding some recently published key developments about cellulose-based aerogel for various applications such as Journal of Materials Chemistry C, 2021, 9, 13103-13114, Composites Part B: Engineering, 2020, 199, 108279, etc.

Author Response

In my opinion, the manuscript can be accepted for publication after adding some recently published key developments about cellulose-based aerogel for various applications such as Journal of Materials Chemistry C, 2021, 9, 13103-13114, Composites Part B: Engineering, 2020, 199, 108279, etc.

Our response: More reference papers related to various applications of cellulose-based material including packaging, straws, foams, flexible electronics, and tissue engineering materials are included in this revised manuscript at line 71– 75 and two papers related to cellulose-based aerogel – Composites Part B: Engineering, 2020, 199, 108279 has been cited as ref. 26 and Journal of Materials Chemistry C, 2021, 9, 13103-13114 has been cited as ref. 29 at the end of line 75.

Reviewer 2 Report

Dear authors, this is a big labor conducted in the field and it is a useful report. I would propose to make more subchapters to include all details, to be more analytical, descriptive. I.e., write more text to be more refreshing. With kind regards

Author Response

Please see the attached file, response to Reviewer 2.

Reviewer 3 Report

The work reviews advances in chemically modified cellulose and its derivates for food packaging applications. It is well known that cellulose is the most widespread polysaccharide in nature. Many derivatives of cellulose are of special interest thanks to their unique properties. The work considers a lot of modern investigations in this field. There are some comments:

Fig. 2: there is a "problem" with chemical formula of cellulose.

Line 69 – indices instead of indexes.

The review contains a lot of abbreviations, and in my opinion, they should be presented in the very beginning in a separate table.

EAB and TS should be substituted for symbols ε and σ.

Tables 3, 4, 7, 8 are overloaded with information.

The reference section can be supplemented with more literature.

Despite these drawbacks, the review concerns lots of recent advances in this field of research. The work requires minor revision.

Author Response

Question 1: Fig. 2: there is a "problem" with chemical formula of cellulose.

Our response: The chemical formula of cellulose has been revised in Figure 2 at line 187 in the revised manuscript by adding “n” at the right corner of the brackets.

Question 2:Line 69 – indices instead of indexes.

Our response: In line 89 in the text of revised manuscript, the “indexes” has been changed to “indices”.

Question 3: The review contains a lot of abbreviations, and in my opinion, they should be presented in the very beginning in a separate table.

Our response: We have supplemented a separate Abbreviation table in the very beginning of the revised manuscript in line 33 in page 2, which contains the nouns which were used in the text over 3 times.

Question 4: EAB and TS should be substituted for symbols ε and σ.

Our response: In the revised manuscript, all the “EAB” and “TS” have all been substituted by symbols “ε” and “σ”. In addition, we also substituted “Young’s modulus by “E”.

Question 5: The reference section can be supplemented with more literature.

Our response: 57 more reference papers have been supplemented in the revised manuscript.

Reviewer 4 Report

A review paper such as this one may eventually lead to highly hydrophobic, cellulosic packaging. This would be a more sustainable alternative to conventional packaging as it would not require depleting resources, polluting the environment, or being sticky. The planet and its inhabitants could benefit from it in the future.

Author Response

A review paper such as this one may eventually lead to highly hydrophobic, cellulosic packaging. This would be a more sustainable alternative to conventional packaging as it would not require depleting resources, polluting the environment, or being sticky. The planet and its inhabitants could benefit from it in the future.

Our response: thanks for the encouraging comments.
